# A 91-Year-Old Female with Recurring Coma Due to Atypical Hyperammonemia

**DOI:** 10.3390/reports8030107

**Published:** 2025-07-14

**Authors:** Manuel Reichert

**Affiliations:** Department of Internal Medicine II, City Hospital (Städtisches Klinikum), 38118 Braunschweig, Germany; m.reichert@skbs.de

**Keywords:** coma, hyperammonemia, metabolic encephalopathy

## Abstract

Background and clinical significance: Acute reduction in vigilance is a frequent reason for emergency department admissions, especially among the elderly. While intracranial causes or infections with fluid depletion are often responsible, there remain cases where imaging, laboratory tests, and clinical examination fail to provide a clear diagnosis. Case presentation: A 91-year-old woman was presented to the emergency department with recurrent episodes of somnolence to deep coma. On admission, her vital signs were stable, and cerebral CT imaging revealed no intracranial pathology. Laboratory analyses, including blood gas measurements, were unremarkable. Empirical treatment for possible intoxications with benzodiazepines or opioids using flumazenil and naloxone had no effect. An Addison’s crisis was considered but excluded following methylprednisolone administration without improvement in consciousness. Eventually, an isolated elevation of serum ammonia was identified as the cause of the reduced vigilance. Further investigation linked the hyperammonemia to abnormal intestinal bacterial colonization, likely due to a prior ureteroenterostomy. There was no evidence of liver dysfunction, thus classifying the condition as non-hepatic hyperammonemia. Therapy was initiated with rifaximin, supported by aggressive laxative regimens. Ammonia levels and vital parameters were closely monitored. The patient’s condition improved gradually, with serum ammonia levels returning to normal and cognitive function fully restored. Conclusions: This case highlights an uncommon cause of coma due to non-hepatic hyperammonemia in the absence of liver disease, emphasizing the diagnostic challenge when standard evaluations are inconclusive. It underscores the need for broad differential thinking in emergency settings and the importance of considering rare metabolic disturbances as potential causes of altered mental status.

## 1. Introduction

This case presentation from a non-university hospital of maximum care in Northern Germany summarizes the initial assessment and further treatment of a 91-year-old female who was admitted unconscious to the emergency department. Here, we present the initial diagnostic and treatment management, ruling out all usual factors of somnolence or coma. The surprising findings and hypothesized cause, hyperammonemia from bacterial overgrowth of an ureteroenterostoma with exemplarly Proteus species after uro-enteral surgery due to bladder cancer, are then revealed. Hyperammonemia was successfully treated with the application of Rifaximin and laxative treatment. Aberrant colonization of both the gut and the urinary tract is common and, in some cases, can lead to a reduction in vigilance. However, the case presented here is a rare situation of surgically altered anatomy resulting in recurring severe infections with subsequent coma. To our knowledge, such a cause of reversible coma has been reported only once, which was published by Skipina and colleagues [1].

## 2. Case Presentation

A 91-year-old female was admitted to the ER after she was found unconscious by her relatives at home. No information about her medical history was available, neither documented nor provided by the relatives. It was reported that despite her age, the patient was independently living in her own home without the need of any help. Documented premedication included Candesartan, Vitamin D, and folic acid. When initially examined, the patient was unconscious with a GCS of 6. All vitals were stable, heart rate was 90/min, blood pressure was 161/81 mmHg, and body temperature was 36.6 °C. Her breathing was 18/min and saturation was 99% without oxygen support. Thus, there was no initial hint of sepsis or an infectious cause of unconsciousness. The pupils were equal and reactive, and no pathologic reflexes could be detected. Her heart sounds were tachycardic but rhythmic without any murmurs, and peripheral pulses were palpable without exceptions. Normal breathing without obstruction or rales could be auscultated. The abdomen was soft and non-tender, and bowel movements were regular in the auscultation. Her skin and mouth were dry but not pale at all. No injuries were found on the extremities. Initial laboratories, including blood gases (Table 1), ECG (Figure 1), chest X-ray (Figure 2), and liver sonography (Figure 3), were obtained. A compensated metabolic acidosis was observed; all other findings were unremarkable as stated.

### Diagnostic Workup

The initial ECG showed a sinus rhythm, left axis deviation, a slightly elevated heart rate of 91/min, and no dysrhythmias or repolarization abnormalities, such as T-wave changes suggestive of potassium imbalance (Figure 1). Also, laboratory chemistry showed no evidence of electrolyte imbalances like hypo- or hypernatremia, which are a common cause of neurological disorders in the elderly (Table 1). However, in the initial blood gas analysis, a compensated metabolic acidosis was detected. Unfortunately, lactate could not be measured due to technical issues. In order to exclude pneumonic infiltrates, and thereby an infectious cause of the acidosis, a chest X-ray was performed, despite initially low CRP levels (Figure 2). There were no signs of venous congestion or pneumonia, the latter being a common cause of altered mental status in elderly patients. No pneumothorax or skeletal injuries were seen. The upper mediastinum was slightly broadened due to a known goiter, but the thyroid parameters appeared unremarkable in the laboratory tests, so we did not associate the goiter with the reduced vigilance. A urine sample showed no evidence of a urinary tract infection.

Since intracerebral events—particularly thromboembolic strokes—are a common cause of comatose states in very elderly patients, a cranial CT (CCT) was performed [2]. Here, no signs of stroke or cranial bleeding could be detected; thus a structural cerebral cause could be neglected as a possibility. In the absence of clinical indicators of epilepsy and due to technical limitations in the emergency department (ED), EEG assessment was postponed Considering other common causes of unconsciousness in particular in the elderly, and given the patient’s unknown medical history, overdose on sedatives could not be neglected for sure. So both 0.4 mg of Naloxone and 0.5 mg of Flumazenil were applied probatorily but no beneficial effect was observed. Hence, an overdose of benzodiazepines or opioids could be ruled out. Although adrenal insufficiency (Addison’s crisis), a potential cause of unconsciousness, seemed unlikely given the stable electrolyte and glucose homeostasis, 250 mg of methylprednisolone was administered intravenously as a last-resort measure due to lack of alternative treatment options in the emergency department. However, no changes regarding consciousness were observed. A cerebrospinal fluid puncture was performed, but neither metabolic imbalance nor signs of an infection were detected. The patient was then transferred to the general internal medicine ward in a cardiopulmonary stable condition, and diagnostic tests were scheduled for the following days.

To exclude a non-convulsive epilepsy state, an EEG was performed In the EEG, Theta-Delta in the sense of sharp waves at alternating locations but no signs of epilepsy could be detected. However, according to the assessment of our neurology colleagues, the EEG suggested an encephalopathy, possibly of metabolic cause. Since all other common causes of coma already had been eliminated, blood alcohol and in particular Ammonia were measured, the latter being one of the leading causes of encephalopathy [3,4,5,6]. The blood test for Ethanol was negative but Ammonia was significantly increased to 346.9 umol/L (normal range 11–51 umol/L). Since the most common cause of hyperammonemia is (alcohol-derived) liver cirrhosis [7,8], an abdominal sonography was then performed but the liver showed no structural impairment (Figure 3). Moreover, transaminases were normal initially. Thus, hepatic encephalopathy HE as a cause could be nullified. Nevertheless, a therapy with 550 mg of Rifaximin twice a day was induced. Additionally, an aggressive laxative therapy with 10 mL of lactulose three times a day and multiple enemas per day was administered and vitals and Ammonia levels were monitored.

Over the next few days, Ammonia levels decreased and the patient’s mental state improved gradually. Finally, she recovered to total awareness and could be interviewed. It turned out that about 20 years prior she had suffered from bladder cancer and a surgical cystectomy and ureteroenterostomy had been performed. Unfortunately, the retention period for medical records in Germany is only 10 years, so no medical reports or similar documents were available. The patient herself had not kept any copies. The care at that time was provided in a hospital at a different location. So, we were confronted with a case of a patient who suffered from deep unconsciousness due to hyperammonemia, where the elevated ammonia levels—unlike in the majority of cases—were not of hepatic origin. Furthermore, a surgical shunt between the intestine and urinary tract had been created. Since there was no other conclusive explanation, we hypothesized pathological microbiota gut colonization and thus, dysbiosis-derived abnormal fermentation with Ammonia by-products. Although *E. coli* is by far the most common cause of urinary tract infections, Proteus mirabilis infections are seen quite often. Proteus species, amongst others such as Klebsiella, Pseudomonas, Staphylococcus, or Morganella, can produce Ammonia from Urea. Given the patient’s surgiucal history and the markedly elevated ammonia level, Proteus species—as part of the normal intestinal microbiota—were considered the most plausible ammonia-producing pathogens responsible for the impaired ammonia levels, particularly since no systemic infection was observed. No alternative explanation was identified; therefore, we postulated bacterial overgrowth from an Ammonia-producing species, such as Proteus. Although we did not perform specific microbiological testing to confirm bacterial colonization, this was due to the constraints of working in a community hospital setting with significant discharge pressure. Given the patient’s clear clinical improvement and the plausibility of our hypothesis regarding dysbiosis and Proteus species involvement, the decision was made to discharge the patient without further microbiological investigations.

It is worth mentioning that over the next months, the patient was admitted several times to the ED with the same symptoms and could be treated successfully each time with the treatment regime described above. Considering the recurring symptoms and elevation of Ammonia, we decided to start a prevention therapy with 200 mg of Rifaximin three times a day. This was already advised, e.g., by Flamm et al. [1], but since such causes of encephalopathy are quite rare, this is not an SOP in German guidelines. Nevertheless, since the start of Rifaximin prevention treatment, the patient did not suffer from hyperammonemia again.

## 3. Discussion

Cases of unclear coma and unconsciousness are seen regularly in emergency medicine departments. The most common causes—amongst intoxications—are cerebrovascular, respiratory, and metabolic imbalances and infections. Regarding a population comparable to Germany, Forsberg and colleagues investigated and demonstrated this in a Swedish ED [2]. In particular, stroke, hypo- and hyperglycemia, hypercapnia, and sepsis are the events seen most often. Those typical etiologies have been demonstrated before [9]. Usually, clinical and neurological examination, CT scans, and standardized laboratory testing including blood gas analysis can reveal the underlying cause quickly and reliably. Relatively rare causes of coma include hormonal insufficiencies such as Addison’s disease or thyrotoxic storm [9]. Kim and colleagues investigated 2028 cases of coma in emergency departments [9]. In addition to the common and rare causes of acute unconsciousness mentioned above, it is notable that in 144 cases (7.1%), the underlying cause of the coma could not be identified. This illustrates that although a definitive diagnosis is always desirable, the reality often differs. This was precisely the case in our patient, where despite extensive diagnostics, no conclusive etiology was found initially. Table 2 summarizes our differential diagnostik steps and conclusion taken from the results. Ultimately, clinical improvement following the initiated therapy was decisive, in line with the principle of diagnosis ex juvantibus.

If the most common causes of unclear coma have been excluded, thinking outside the box is necessary [4,10]. In the case presented here, a combination of infectious and metabolic issues due to gut dysbiosis with hyperammonemia was hypothesised to be the underlying cause of coma. By far, the most common etiology of hyperammonemia is liver failure, mostly due to alcohol-derived cirrhosis. This can be easily detected by sonography. From the case presented here, we conclude that in cases of unclear consciousness after the elimination of typical dysfunctions, Ammonia should be measured even if sonography of the liver shows no abnormal structures. Cases of hyperammonemia due to urinary tract abnormalities such as urine retention—especially in children—have been reported but remain rare. To our knowledge, to date only one case of hyperammonemic encephalopathy in patients with surgical impairment of the urinary tract has been published so far. Skipina et al. reported a patient with ileocecal pouch after cystectomy with hyperammonemia [1]. As far as our research shows, hyperammonemia caused by gastrointestinal and urinary tract infections with ammonia-producing bacteria remains a rare but conclusive scenario, which we consider the most plausible explanation in the present case.

## 4. Conclusions

The case presented here is another rare example of hyperammonemia due to the surgical connection between the gut and urinary tract. Given the excellent treatment response to laxatives and rifaximin, dysbiosis of microbiota should be considered the most plausible reason for the symptoms described above and prophylaxis could be applied in patients after uroentero-connective surgery, such as performed in the case presented here. This case also stresses the importance of out-of-the-box-thinking. A German saying among medical professionals states “What is common is common.” The case presented here illustrates how misleading such an approach can be.

## Figures and Tables

**Figure 1 reports-08-00107-f001:**
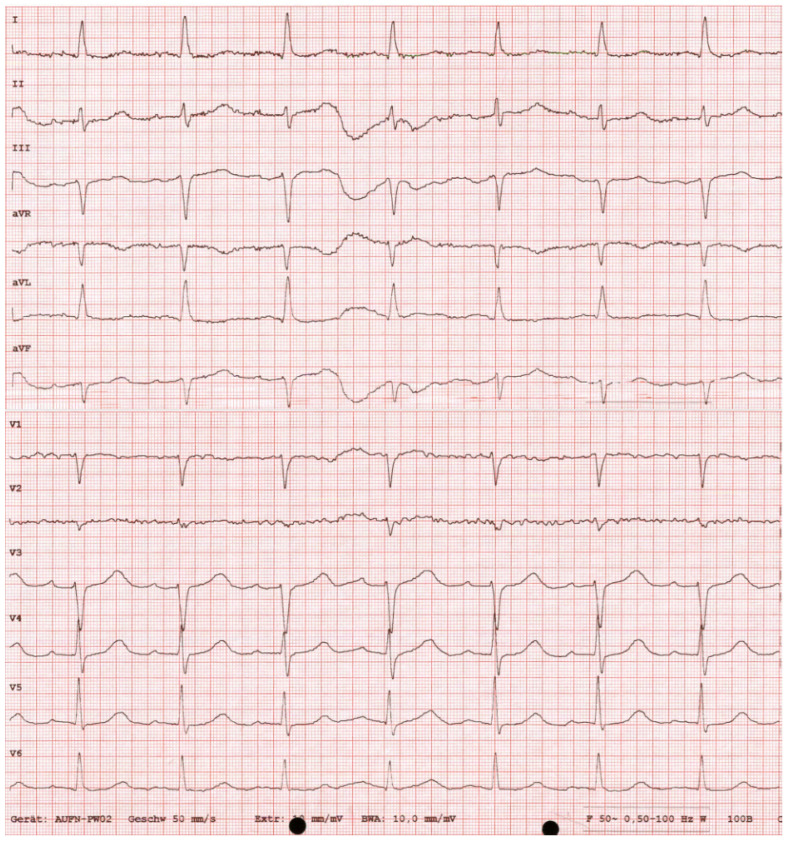
The No scientific information is lost, since all necessary waveform morphology and intervals can be assessed visually using the standard scaling (1 mm = 40 ms). For readers outside of cardiology, these values are of limited interpretative relevance. Therefore, I believe that the figure remains scientifically complete and appropriate as presented. ECG at admission showed no severe dysrhythmias nor signs of typical intoxications. I–III represent the standard limb leads; aVR, aVL, and aVF are augmented limb leads; V1–V6 are the precordial (chest) leads.

**Figure 2 reports-08-00107-f002:**
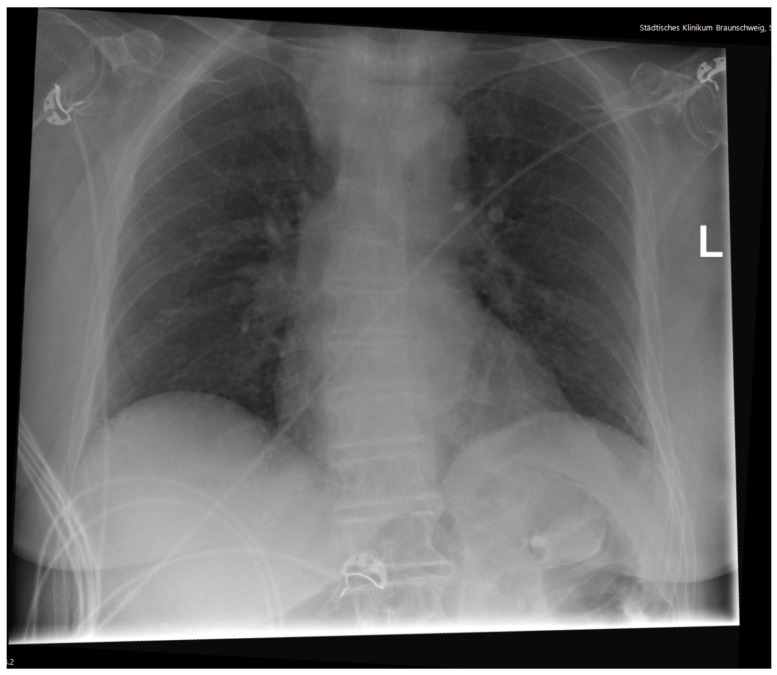
Chest X-ray showed no pneumonia or sign of heart failure. Pneumothorax or pleural effusions were not visible either. The upper mediastinum was slightly broadened due to a known goiter. The letter “L” denotes the patient’s left side and serves as a standard orientation marker in posteroanterior chest radiography.

**Figure 3 reports-08-00107-f003:**
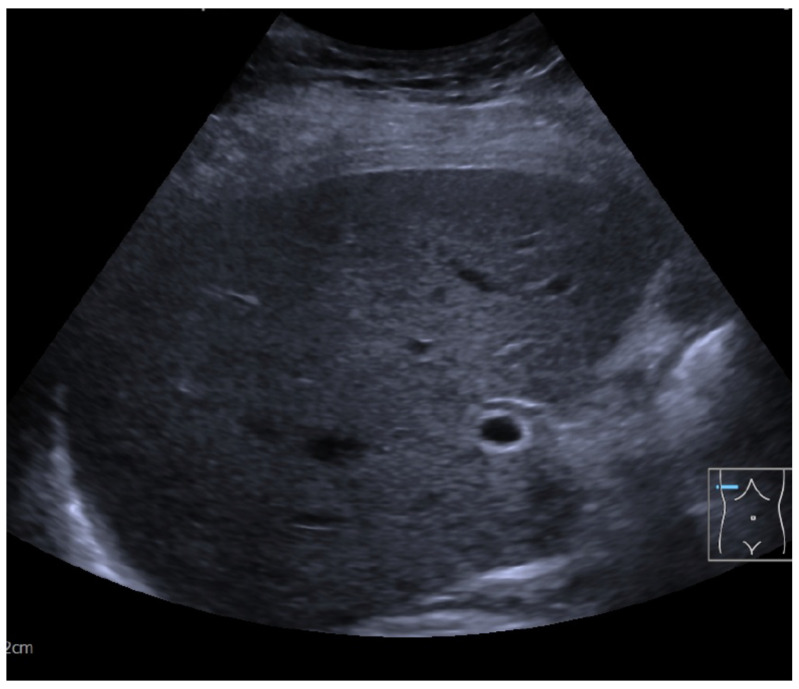
In the sonographic liver examination, no cirrhosis was detectable. The blue line indicates the orientation marker on the ultrasound probe, placed transversely in the right upper abdominal quadrant.

**Table 1 reports-08-00107-t001:** Initial blood testing and blood gas analysis.

Laboratory Test		Result	Normal Range
Blood count			
	Red blood count [M/mcL]	4.44	3.93–5.22
	White blood count [K/mcL]	5.8	3.98–10.04
	Hemoglobin [g/dL]	13.2	11.2–15.7
	Hematocrit %	38.1	34.1–44.9
	Platelets [K/mcL]	295	160–370
Chemistry			
	CRP [mg/L]	1.5	<5
	Sodium [mmol/L]	144	136–145
	Potassium [mmol/L]	3.95	3.4–4.5
	Calcium [mmol/L]	2.31	2.20–2.55
	Magnesium [mmol/L]	0.79	0.66–0.99
	Creatinine [mg/dL]	0.68	0.51–0.95
	Bilirubin total [mg/dL]	0.5	<1.20
	Alanine transaminase [U/L]	26	<33
	Aspartate transaminase [U/L]	31	10–35
	Glucose [mg/dL]	124	75–128
	Cardiac troponin T [pg/mL]	28	<14
Arterial blood gas			
	Arterial carbon dioxide [mmHg]	22.3	35–45
	Arterial oxygen [mmHg]	54.3	75–97
	pH	7.379	7.35–7.45
	Base excess [mmol/L]	−9.8	−3–2
	HCO_3_^−^ [mmol/L]	13.2	20–26

M = 10^6^; K = 10^3^; mcL = microliters; mg = milligrams; dL = deciliters.

**Table 2 reports-08-00107-t002:** Overview of differential diagnoses and reasons for exclusion. Double upward arrows (↑↑) indicate a significantly increased value compared to the reference range.

Differential Diagnosis	Consideration	Tests Performed	Results
Infection	Most frequent cause in the elderly	Leukocytes, CRP, vitals	No evidence of infection
Metabolic disturbances	Common cause of altered consciousness	Glucose, electrolytes, pH, BE, ${\mathrm{HCO}}_{3}^{-}$, EtOH, NH_3_	NH_3_ ↑↑
Neurological events	Very frequent cause in the elderly	Examination, CCT, spinal fluid, EEG	Suspected encephalopathy
Toxic ingestion	Possible, often overdose of sedatives	Clinical examination, antidotes administered	Not suspected
Endocrine disorders	Rare but possible	Electrolytes, ECG, lab testing, steroid administered	Not suspected

## Data Availability

The original contributions presented in this study are included in the article. Further inquiries can be directed to the corresponding author. No additional data of relevance beyond those presented in the article are available.

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
