# Peer review of "A 91-Year-Old Female with Recurring Coma Due to Atypical Hyperammonemia"

_reports, 2025, doi:10.3390/reports8030107_

Round 1
Reviewer 1 Report
Comments and Suggestions for Authors
The author tried to describe a 91-year-old female with recurring coma upon atypical hyperammonemia.
Although the author has attempted to describe this case report in detail, the choice of diagnostic procedures should be better explained. The patient's treatment should also be more clearly explained. A more detailed explanation and critical assessment of the diagnostic steps taken, and the therapy applied.
A more detailed discussion of the differential diagnoses is needed.
When exactly was the bladder tumor surgery performed, which is speculatively cited as the cause of the coma. You must justify your claim more appropriately and with more evidence (list references).
Reviewer 2 Report
Comments and Suggestions for Authors
Manuel Reichert described a case of a patient with recurring coma due to hyperammonemia.
I agree with the author that the presented case is not typical regarding the reasons for hyperammonemia. I do understand that she had undergone cystectomy and ureteroenterostomy and therefore, the link between the kidneys and gut was established. In normal conditions, urea is aseptic, and Proteus spp. are part of the typical human microbiome.
My questions are:
- What is the supposed reason for the increased Proteus prevalence in the gut?
- Whether spontaneous bacterial overgrowth (SIBO) or dysbiosis was investigated, would enhance the plausibility of the proposed mechanism.
- Table 1 shows a compensated metabolic acidosis (base excess -9.8 mmol/L, HCO₃⁻ = 13.2 mmol/L, pH 7.379), which is not mentioned in the main text. The CRP level is low, and no signs of sepsis are noted, so this acid-base disturbance warrants discussion. Was lactic acid measured? Could this reflect subclinical ischemia or catabolic state?
Minor issues:
- Figures 1 and 2 both refer to the ECG, which seems redundant, especially since there is a Chest X-ray in Figure 2.
English language:
- “miscolonization”: it seems that “aberrant colonization” or “bacterial overgrowth” could be better,
- “could be ruled out” is often repeated: rephrasing to avoid redundancy might improve the text,
- "isochor reaction": I wonder if “isocoric reaction” or simply “pupils equal and reactive” would not be a better choice.
Round 2
Reviewer 1 Report
Comments and Suggestions for Authors
I have nothing more to add. Thank you for the answers.